# Application of Dual-Enhanced Surface-Enhanced Raman Scattering Probe Technology in the Diagnosis of Tumor Cells in Vitro

**DOI:** 10.3390/molecules27113582

**Published:** 2022-06-02

**Authors:** Yinping Zhao, Yawei Kong, Liwen Chen, Han Sheng, Yiyan Fei, Lan Mi, Bei Li, Jiong Ma

**Affiliations:** 1Institute of Biomedical Engineering and Technology, Academy for Engineering and Technology, Fudan University, Shanghai 200433, China; 19110860079@fudan.edu.cn (Y.Z.); 19110860077@fudan.edu.cn (H.S.); 2Department of Optical Science and Engineering, School of Information Science and Technology, Fudan University, Shanghai 200433, China; 18110720005@fudan.edu.cn (Y.K.); 21110720090@m.fudan.edu.cn (L.C.); fyy@fudan.edu.cn (Y.F.); 3State Key Laboratory of Applied Optics, Changchun Institute of Optics, Fine Mechanics and Physics, Changchun 130000, China; 4Shanghai Engineering Research Center of Industrial Microorganisms, The Multiscale Research Institute of Complex Systems (MRICS), School of Life Sciences, Fudan University, Shanghai 200433, China

**Keywords:** SERS probe, dual enhancement, stable and highly sensitive, Raman signals, A549 cell

## Abstract

With the development of precision medicine, antigen/antibody-targeted therapy has brought great hope to tumor patients; however, the migration of tumor cells, especially a small number of cells flowing into blood or other tissues, remains a clinical challenge. In particular, it is difficult to use functional gold nanomaterials for targeted clinical tumor diagnosis while simultaneously obtaining stable and highly sensitive Raman signals. Therefore, we developed a detection method for functional Au Nanostars (AuNSs) with dual signal enhancement that can specifically track location and obtain high-intensity surface-enhanced Raman scattering (SERS) signals. First, AuNSs with specific optical properties were synthesized and functionalized. The Raman dye 4-mercapto-hydroxybenzoic acid and polyethylene glycol were coupled with the tumor marker, epidermal growth factor receptor, to obtain the targeted SERS probes. In addition, a detection chip was prepared for Raman detection with physical enhancement, exhibiting a 40-times higher signal intensity than that of quartz glass. This study combines physical enhancement and SERS enhancement technologies to achieve dual enhancement, enabling the detection of a highly sensitive and stable Raman signal; this has potential clinical value for antigen/antibody-targeted tumor diagnosis and treatment.

## 1. Introduction

Cancer is a major health concern worldwide. According to recent statistics, 606,880 Americans died of cancer in 2019, which translates to nearly 1700 deaths per day [1]. At present, the treatment of tumors is focused on three traditional treatment strategies: surgery, chemotherapy, and radiotherapy. Approximately 60% of tumors are treated with surgery; however, it has inevitable adverse effects on the patient’s body [2]. In addition, the migration of tumors, especially of a small number of cells flowing into the blood or other tissues, cannot be overcome [3,4]. The development of biotechnology and the in-depth study of pathogenesis at the cellular and molecular levels have ushered in a new era for tumor therapy with molecular-targeted treatment attracting substantial attention. Molecular targeted therapy for tumor involves the use of specific carriers to transport tumor drugs or other substances that kill tumor cells to the tumor site in a direct and oriented manner [5]. When compared with the traditional methods of surgery, radiotherapy, and chemotherapy, molecular-targeted therapy shows the most thorough “curative” effect and can further reduce the associated pain and economic burden for patients [6]. Most molecular-targeted therapeutic drugs are monoclonal antibodies against specific tumor cell markers, tumor cell signal transduction inhibitors, anti-angiogenesis drugs, and antagonists of certain cytogenetic markers or oncogenes [7]. Epidermal growth factor receptor (EGFR) is closely related to cell differentiation, proliferation, and tumor metastasis and is involved in the proliferation, invasion, metastasis, and angiogenesis of many types of tumors [8]. EGFR mutations are common in many cancers, including lung, breast, cervical, esophageal, and gastric cancers. Therefore, EGFR is a key target for precise targeted therapy of various tumors [9,10,11,12,13].

A modified gold (Au) nanoprobe for antibody binding is a valuable tool for tumor-targeted therapy owing to the optical properties of Au nanoparticles and the advantages of the nanoparticle size, their easy preparation, easy modification, and low cytotoxicity [12,14,15,16,17]. When compared to other Au nanoparticles (gold balls and rods), Au Nanostars (AuNSs) have five sharp corners, which can easily produce tip effects, strong electric fields, and an enhancement effect [18]. Polyethylene glycol (PEG) coupled to the surface of Au nanoparticles can enter the cell through endocytosis. This prolongs the circulation time of Au nanomaterials in the blood and increases biocompatibility [19]. These developments have expanded the application of nano-Au in biology. The Raman spectrum is generated by chemical molecular vibrations, which can identify the internal molecular structure of substances; however, the Raman signal is generally weak. Therefore, the routine application of Raman spectroscopy is generally limited in the field of bioscience, especially at the micron level, as the volume of a single cell is relatively small and the content of some proteins and other substances is extremely low. Alternatively, surface-enhanced Raman scattering (SERS) is commonly used to generate high-intensity and specific signals for targeting and single-molecule detection [3,11,17,20,21]. SERS is applicable to common colloidal metal particles used as substrates, such as metal samples with rough surfaces, including copper, silver, and gold. These rough samples are adsorbed or modified to produce strong Raman spectral signals. Chen et al. identified clear surface plasmon resonance between interacting Au nanoparticles [11], mainly because there are many suspended parts on the surface of the nanoparticles that can easily react with other groups. In other words, it is easy to carry out surface functional modifications on Au nanoparticles. When the incident light interacts with the free electrons on the surface of the modified functional group, the Raman signal can be enhanced; this SERS signal could aid in the detection of trace substances, making it suitable for the diagnosis and targeted treatment of early tumors [22,23,24].

The complex mechanisms of tumor cell invasion and growth require a special tumor microenvironment to meet specific growth needs, including the pH and calcium concentration; in addition, tumor cells have a strong migration capability [25,26]. Traditional surgery, radiotherapy, and chemotherapy cannot overcome the migration of tumors, especially the small number of cells flowing into the blood or other tissues [27,28]. Therefore, tracking tumor cell movement based on the target and location and obtaining stable and sensitive Raman detection signals simultaneously remains one of the main challenges in the application of functional Au nanomaterials in clinical tumor-targeted diagnosis. To address this problem of signal sensitivity, we developed a detection method that can realize specific localization, tracking, and a high-intensity SERS signal, which was applied to the diagnosis of tumor cells in vitro [21]. First, a functionalized Au nanomaterial was prepared. Surface-modified Au nanoparticles were combined with the EGFR of tumor cells to enable targeted localization [20,29,30]. The signal from the coupling of antigen and antibody was detected using surface-enhanced Raman technology, obtaining high sensitivity and high-intensity SERS signals of the functionally modified AuNSs. In addition, a new detection chip was prepared for the collection of Raman detection signals. This chip can physically enhance the Raman signal of functionalized nanoparticles, with a signal intensity that is 40 times higher than that of quartz glass. This method combines physical enhancement and SERS enhancement technology to achieve a dual-enhancement effect to enable the detection of tumor cells with high sensitivity and stability. This highly stable and sensitive functional material has potential clinical value for the diagnosis of tumor cells (Figure 1.).

## 2. Material and Methods

Chloroauric acid (HAuCl4), 4-hydroxyethyl piperazine ethyl sulfonic acid (HEPES), mercapto-polyethylene glycol (PEG) with a molecular weight of 5000, and p-mercaptobenzoic acid (MBA) were purchased from Shanghai Aladdin Reagent Company (Shanghai, China). Anti-epidermal growth factor receptor (EGFR) was purchased from Abcam. The cell line (A549) was purchased from ATCC (Shanghai, China). DMEM and fetal bovine serum were from GIBCO; CCK-8 kit, was from Promega; L929 cells were from ATCC (Shanghai, China). All the reagents were analytically pure.

### 2.1. Synthesis of AuNSs

The weight of 4.766 g was determined using an analytical balance in a beaker after drying. Two-hundred microliters of Ultrapure water was added to prepare the lipopolysaccharide solution, which was evenly stirred. The pH of the HELPS solution was adjusted to 7.4, and 400 μL HAuCl_4_ (0.1 M) was added and allowed to stand at room temperature for 2 h. The reaction was stopped when the color of the solution changed from blue to black. The mixture was centrifuged at 14,000 rpm for 15 min, and the supernatant was removed. The precipitate was dispersed in sterilized water using ultrasonication. The washing steps were repeated two to three times. Excess HELPS solution was removed, and the reaction was stopped [31]. The solution was then stored until the characterization of the synthetic morphology using electron microscopy and ultraviolet (UV) absorption measurements.

### 2.2. Surface Functional Modification and Optimization of AuNSs

Three microliters of the Raman dye 4-mercapto-hydroxybenzoic acid (MBA) solution (50 μM) was added to 1.8 mL AuNSs solution in a constant-temperature oscillation reaction overnight to form Au-S [31]. When the reaction was complete, the solution was centrifuged at 14,000 rpm for 15 min, carefully washed twice with deionized water to remove the residual supernatant, and the precipitate was dispersed in 1 mL of sterilized water through ultrasonication.

Subsequently, 20 μL PEG (20 mM) solution was added and the solution was incubated overnight under constant temperature oscillation, which improves stability and protection [19]. When the reaction was complete, the mixture was centrifuged at 14,000 rpm for 15 min, washed with deionized water three times, the supernatant was removed carefully, and the precipitate was dispersed in sterilized water through ultrasonication. Following the centrifugation (twice), the excess MBA and PEG solution was removed, and the precipitate was reconstituted in 1.8 mL sterilized water. Finally, the EGFR antibody was added and allowed to react overnight, and the above cleaning steps were repeated. Solutions with pH values of 5, 6, and 7 were prepared in advance with ultra-pure water, which was aseptically filtered on an ultra-purification table with a sterile syringe and a 0.22 μm filter membrane. The functional nanomaterials were dispersed evenly in sterile solutions with pH values of 5, 6, and 7, and set aside.

### 2.3. Cytotoxicity Test

Logarithmic growth phase cells (L929) were plated in a 96-well plate, at a density of 1 × 10^4^ cells/well, for adherent culture. Wells containing only the culture medium served as the blank control group. The experimental control group included cells with culture medium after adherent culture, to which functional Au nanomaterials were added. Each group was established using six replicates. After incubation for 24 h, 10 μL of the CCK-8 solution was added to each well and incubated for 2–4 h. The absorbance was measured using an enzyme-labeling instrument, and the cell survival rate was calculated.

### 2.4. Raman Experiment

#### 2.4.1. Surface-Functionalized AuNSs for Raman Detection of Cells in Vitro

After the functional nanomaterials were incubated with human lung adenocarcinoma cells (A549) for 24 h, the culture medium was removed and the cells were washed with phosphate-buffered saline three times. The cells were digested with trypsin digestion and centrifuged at 800 rpm for 3 min, washing three times with PBS buffer solution, the supernatant was removed, and the cells were fixed for more than 4 h with cell fixing solution; 2 µL of the sample was placed on a Raman chip for detection with a power of 3 mW and an integration time of 10 s. Single cells were scanned using regional mapping [32]. Raman spectra were measured using a confocal micro-Raman spectrometry system equipped with a cooled semiconductor (−75 °C; Power Integrations, Inc., New Jersey, NJ, USA) and a charge-coupled device detector (1340 × 100 pixels) to achieve a high signal-to-noise ratio (quantum efficiency > 90% at 550 nm). Sample excitation was performed at 532 nm using a fiber-coupled solid-state laser with a 600 g/mm grating, and a 100× objective lens was used to collect the spectra from 285 to 3745 cm^−1^ with a spectral resolution of 3 cm^−1^.

#### 2.4.2. Raman Chip

For Raman detection, the detection substrate used was a Raman chip with a low background and a physical enhancement effect, using 2 µL of the surface-functionalized nanomaterial solution. The surface of the Raman chip was polished and subjected to single-layer metal vacuum evaporation. The thickness of the chip was 100 ± 2 nm, and the surface was smooth and clean without any dust or other dirt. For detection, a 532 nm laser was used with a power of 3 mW, a laser spot size of 300 nm, and an integration time of 10 s. Raman signals of the cells on the Raman chip and quartz slide were collected for comparison.

## 3. Results and Discussion

### 3.1. Characterization of AuNSs

The size and appearance of the AuNSs were relatively uniform; the morphology of AuNSs was pentagonal, further optimizing can obtain a more uniform morphology and size, as shown in Figure 2a. Figure 2b showed that the UV characteristic absorption peaks of AuNSs appeared at approximately 530 and 780 nm. The cytotoxicity verification experiment using the mouse epithelial fibroblast cell line L929 showed that AuNSs have good biocompatibility, with a cell survival rate about 85% after incubation with 1 µg/µL AuNSs (Figure 2c). Au nanoparticles are a metal element without other atomic and molecular chemical bonds, therefore, AuNSs have no Raman signal (Figure 2d).

### 3.2. Synthesis and Characterization of Functional Nanomaterials

After the reaction between the AuNSs and the modified group [33], we preliminarily optimized the reaction ratio and tested the UV absorption of each step of the reaction: AuNSs, AuNSs and MBA, AuNSs, MBA, and PEG, and AuNSs, MBA, PEG, and EGFR. As shown in Figure 3a, the corresponding absorption peaks redshifted after each step of the reaction [34,35]. As shown in Figure 3b, there was no Raman peak when only AuNSs were present, whereas the AuNSs modified with MBA and PEG had strong Raman absorption characteristic peaks at 1084 and 1585 cm^−1^, indicating that the AuNSs modified with MBA and PEG had a strong surface-enhanced Raman effect after surface functionalization. Moreover, the signal of AuNSs modified with MBA and PEG reached a higher enhancement after EGFR modification. The functionalized AuNSs have a good biocompatibility, as shown in Figure 3c. In addition, the pH of the functionalized AuNSs was optimized, as shown in Figure 3d, and the Raman signal of the functionalized AuNSs was confirmed to be stable in an acid-base environment with a pH range of 5 to 7, and was not readily affected by the environmental pH. Therefore, this functional nanomaterial is promising for the targeted tracking and detection of tumor cells in vitro.

### 3.3. Enhancement of the Physical Signals of Functional Nanomaterials

After the functional nanomaterials were incubated with the cells, the samples were placed on a Raman chip and a quartz glass slide for Raman signal acquisition and comparison, using the average of 30 spectra of 10 cells. As shown in Figure 4a, the relative intensity of the signal on the quartz slide was approximately 3 × 10^2^ accounts, because of SERS enhancement. As shown in Figure 4b, the relative intensity of the Raman signal collected on the Raman chip was approximately 1.2 × 10^4^ accounts, which was approximately 40 times stronger than that of the quartz glass, and weaker Raman peaks could be detected. One of the reasons for the physical enhancement of the Raman chip is the superior collection effect, because Raman is a scattering signal. A large part of the scattering signals emitted to the surroundings will be lost because they cannot be collected in the objective lens, whereas ordinary quartz glass has a high transmittance; therefore, a large part of the signals will be lost through the bottom. Similarly, signals around the left and right sides were lost. The Raman chip has a uniform coating and a good reflection effect; therefore, some of these reflected signals excite the sample for the second time and scatter more Raman signals. Simultaneously, these scattered signals were directly concentrated and reflected to the objective lens for collection. Therefore, the signal of the Raman chip was approximately 40 times stronger than that of the quartz glass slide, indicating its superior suitability for the detection of targeted tumor cells in vitro, achieving high sensitivity and targeted weak signals at the same time. Figure 4c shows the Raman data of ordinary quartz and the Raman chip, and Figure 4d shows a comparison of the Raman data between the ordinary quartz and Raman chip with histograms.

### 3.4. Functional Nanomaterials for In Vitro Cell Detection

Functionally modified AuNSs and unmodified AuNSs were added to the cells and dark-field imaging was performed after incubation. As shown in Figure 5a, the cells with unmodified AuNSs showed a certain scattering signal, but they were very weak. The scattering signal intensity of the functionalized AuNSs was strong and most of the scattering signal intensity concentrated around the edge of the cell, while the scattering signal intensity in the middle of the cell is very weak and the imaging is very dark, further confirming the targeting ability of the functionalized AuNSs, as shown in Figure 5b. A Raman chip was used as the detection substrate to collect the Raman signal of a single cell and the cell was scanned by regional Raman mapping, which more intuitively showed the distribution of Raman signal in a cell, as shown in Figure 5c. The color change from black to red corresponds to the change in the Raman signal from weak to strong, as shown in Figure 5d. Figure 5e shows the Raman data comparison; the functionalized AuNSs were incubated with the cells, resulting in a high-intensity Raman signal, whereas there was only a weak Raman signal of the cell itself after being incubated with unmodified AuNSs, and there was no signal enhancement, obtaining high sensitivity and high-intensity SERS signals of the functionally modified AuNSs [34,36], as shown in the Figure 5e.

## 4. Conclusions

In this study, SERS probes were prepared using surface−enhanced functionalized Au nanoparticles. The MBA and PEG with a sulfhydryl group were coupled via an Au–S bond reaction. The surface-modified AuNPs were then combined with the tumor cell marker EGFR to achieve targeted localization to A549 cells. The signal from the coupling of the antigen and antibody was detected using surface-enhanced Raman technology to obtain the targeted and specific Raman signals of A549 cells; therefore, the prepared nanomaterial can specifically identify and track A549 cells. PEG can inhibit both the aggregation and the adsorption of blood serum proteins on its surface, reduce their uptake by the liver, and consequently extend the circulation time, eliminate the non-specific particle-cell binding, as well as can improve the biocompatibility of the SERS probe [19]. 4−Mercaptobenzoic acid (4−MBA) was used as the Raman reporter molecule, which has two Raman characteristic peaks with unselective signal enhancement [36].

To address the problem of signal sensitivity, a coated detection chip was prepared, which exhibits strong reflection and serves as a convergence signal and secondary excitation sample for the SERS probe. The experimental results showed that the relative Raman signal value of ordinary SERS was enhanced to (3 × 10^2^ accounts); the relative Raman signal value of Raman chip was (1.2 × 10^4^ accounts), which is 40 times higher than that obtained on an ordinary quartz slide. This study combined physical enhancement and SERS enhancement technologies to achieve dual enhancement, enabling the detection of a highly sensitive and stable Raman signal. The targeting and specificity of functionalized AuNSs for cancer cells, referenced by the results of previous studies, provides theoretical support for this view [18,36]. If EGFR is replaced by other marker molecules, the same double enhancement effect would be achieved.

In conclusion, we developed a new detection method applied to the diagnosis of tumor cells in vitro that can enable high-sensitivity and high-intensity SERS signals. This highly stable and highly sensitive signal is a very promising detection method for the diagnosis and treatment of tumors in vitro. With continued in-depth research, these nanomaterials could have further applications.

## Figures and Tables

**Figure 1 molecules-27-03582-f001:**
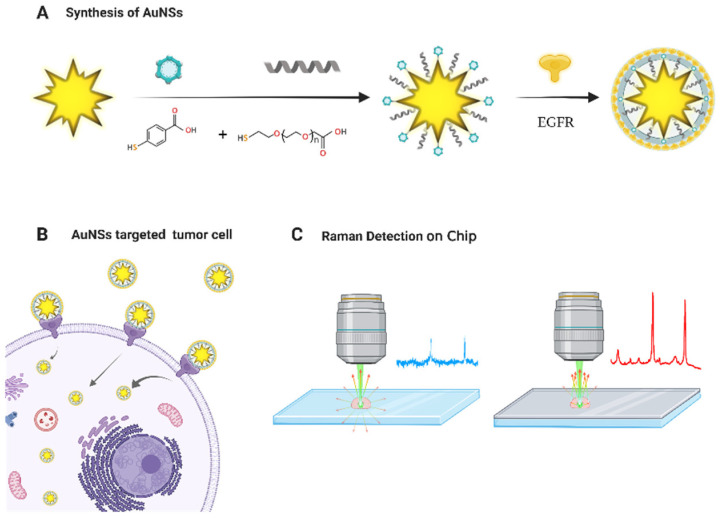
Schematic illustration of the dual-enhanced SERS probe technology in the diagnosis of tumor cells in vitro. (**A**) synthesis od AuNSs; (**B**) AuNSs targeted tumor cell; (**C**) Raman detection on chip.

**Figure 2 molecules-27-03582-f002:**
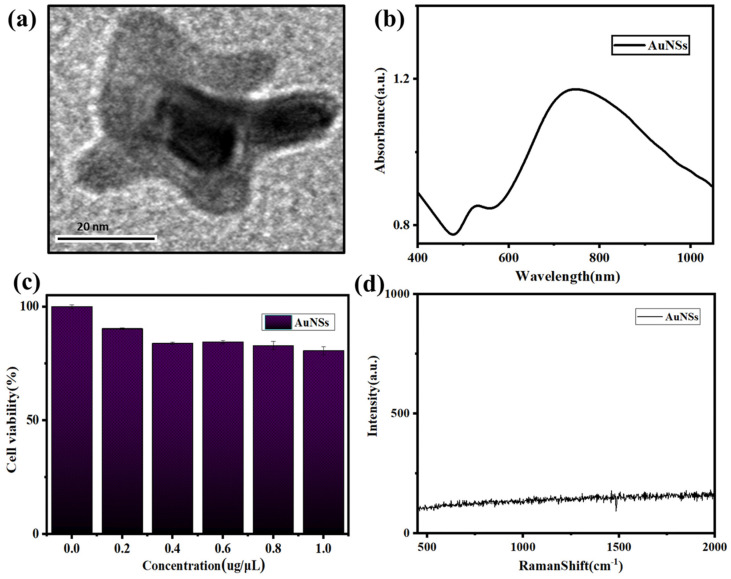
Characterization of the AuNSs. (**a**) Scanning electron microscope images of AuNSs. (**b**) UV-visible absorption spectrum of AuNSs. (**c**) Cytotoxicity test of AuNSs. (**d**) Raman signal of AuNSs.

**Figure 3 molecules-27-03582-f003:**
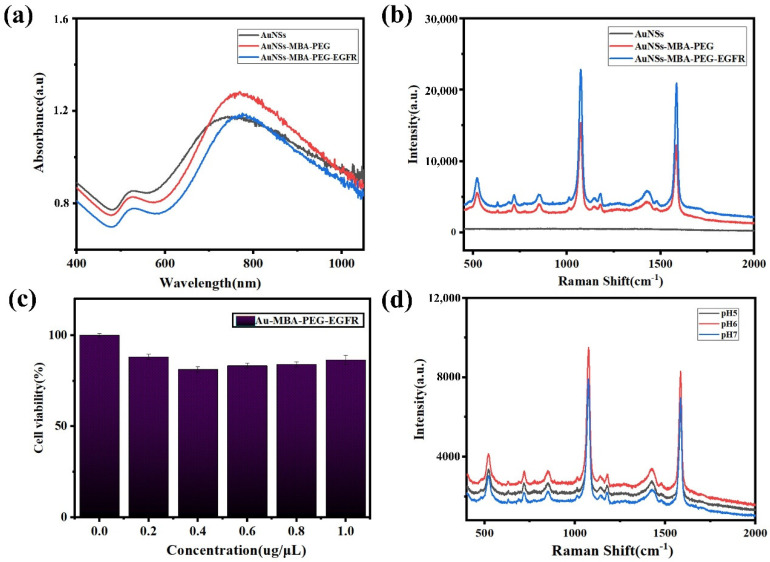
Synthesis and characterization of functional nanomaterials. (**a**) UV−visible absorption spectra of AuNSs, 4−MBA−labeled AuNSs, PEG−4−MBA−labeled AuNSs, and EGFR−PEG−4−MBA−labeled AuNSs. (**b**) Raman spectra of AuNSs, PEG−4−MBA−labeled AuNSs, and EGFR−PEG−4−MBA−labeled AuNSs. (**c**) Cytotoxicity test of EGFR−PEG−4−MBA−labeled AuNSs. (**d**) pH value optimization.

**Figure 4 molecules-27-03582-f004:**
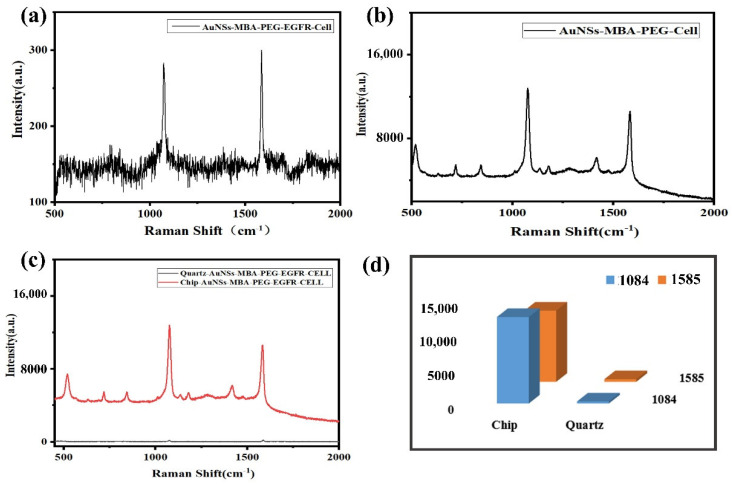
SERS spectra of A549 cells with different detection substrates (**a**) Raman spectra of Cell−EGFR−PEG−4−MBA−labeled AuNSs on the quartz slide. (**b**) Raman spectra of Cell−EGFR−PEG−4−MBA−labeled AuNSs on the Raman chip. (**c**) Comparison of Raman signals on the Raman chip and quartz slide. (**d**) Raman peaks data histogram at 1084 and 1585 cm^−1^ of the Raman chip and quartz slide.

**Figure 5 molecules-27-03582-f005:**
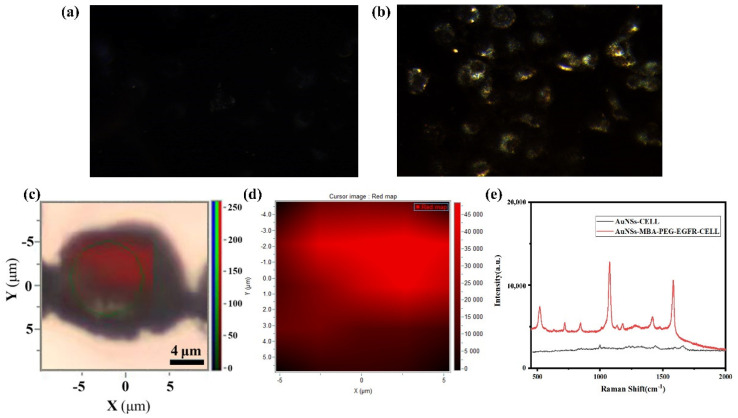
Dark field imaging and SERS imaging of A549 cells. (**a**) Dark field imaging of AuNSs. (**b**) Dark field imaging of EGFR−PEG−4−MBA−labeled AuNSs used the Raman chip. (**c**) Raman mapping of EGFR−PEG−4−MBA−labeled AuNSs. (**d**) The color diagram of the Raman signal intensity. (**e**) Raman data diagram of AuNSs and EGFR−PEG−4−MBA−labeled AuNSs with A549 cells.

## Data Availability

All data supporting the findings of this study are available from the corresponding author upon reasonable request.

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
