# Peer review of "Application of Dual-Enhanced Surface-Enhanced Raman Scattering Probe Technology in the Diagnosis of Tumor Cells in Vitro"

_molecules, 2022, doi:10.3390/molecules27113582_

Round 1
Reviewer 1 Report
Zhao et al present a study in which Au nanostars are functionalized in order to interact with cancer cells. The SERS signals associated to the nanostructures are used as cell labels. Labeling cancer cells with plasmonic nanostructure for SERS detection is of wide interest, several excellent examples can be found in recent literature. Here a couple of examples along others:
- DOI: 10.1039/C8TB02828A
-
DOI: 10.1038/s41598-020-72911-w
The field is therefore well covered, but further investigation is indeed needed in order to extend the proof-of-concept of labeling cells with metal nanoparticles to real practical applications. To do that, one must take in mind that cell must be counted and proper experiments about selectivity must be shown. In this terms, the study of Zhano et al lacks in several aspects. The most important are resumed in the following.
First, the descriptions in the Methods (pag 4, 2.4.1) and Results&Discussion sections are ambiguous. In the first, it seems that the cells are digested. "digestive" and "fixing" solutions must be properly specified. "Digestion" is supposed to drive to cell lysis, while in the Discussion it seems the cells maintain their own structure.
Labeling efficacy must be statistically proved. It means that, at very least, few tens of cells must be measured.
Labeling selectivity must be evaluated. It means using two types of cell lines (one positive to EGFR and one negative) to be both treated with the nanostructures. Alternatively, one may use a single cell line and nanostructures with and without EGFR. The better is, of course, both of the two experiments. Figure 4e does not answer the question, as far as the black trace in labeled as "AuNS - cell". Does it mean that these particles do not have nor MBA, nor PEG and EGFR?
Please provide a better self-revision before submission. Nanostars becomes nano Venus, Figure 1 is missing, caption of Figure 4 does not represent the real figure content. Figure 4a, c and d are of so poor quality to not been actually useful. Bibliography is reported twice. And so on...
For all of the aspects cited above, my recommendation is that the manuscript needs extensive revision before to be re-considered for publication.
Author Response
Please see the attachment,thank you very much !

Reviewer 2 Report
1. Figure-1, is missing, see line no 198.
2. Author should clarify, in figure 2c, why cell viability increases with increasing concentration after 0.4.
3. In methodology, should include the effects under different pH of AuNSs .
4. " In addition, the pH of the material was optimized, as 211
shown in Fig. 2c. " See line no 211, in caption of Figure 2c mentioned "(c) Cytotoxicity test of EGFR-PEG-4-MBA- 219 labeled AuNSs". Both are different.
5. Need to include one comparative table like this-work advantages with other prior works.
Author Response
Please see the attachment, thank you very much !

Round 2
Reviewer 1 Report
The Authors carefully addressed almost all the suggestions, even if the "Original Commet 3", following the Authors Letter list, still remains unsolved. Unfortunately, this is of fundamental importance. The overall study foresees that EGFR functionalized AuNS may selectively label cancer cells and that the associated SERS signal may be used to detect these cells. The MBA signal is therefore the indirect figure of merit to asses that a specific cell is from a cancer or not. This may work only if one trusts the selective binding of the AuNS to the cancer cell, if not one is going to have false positives. Figure 1d and Figure 2b show that AuNS alone do not have any signal to be detected. So that the "negative control" of Figure 4e is not fair. How can one asses if a cell contains or not AuNS if there are not any signal that can prove its presence? The correct experiment, as already mentioned, should had been AuNS-MBA-PEG with cancer cells. So that the absent of signals definitely prove that there are not any stars over there. One may understand that the last years were much more difficult because of COVID, so one does not pretend so much replicates etc, but unfortunately this issue must be considered one of the fundamental evidence about the consinstency of the overall paper. I would not suggest to go ahead with the present interpretation of this specific aspect, or at least without removing that part totally and clearly state in the text that the study lacks in verifying if the AuNS-MBA-PEG-EGFR are selective or not.
Author Response
Please see the attachment,thanks again for your advice!
